# Deep Ensemble Learning Approaches in Healthcare to Enhance the Prediction and Diagnosing Performance: The Workflows, Deployments, and Surveys on the Statistical, Image-Based, and Sequential Datasets

**DOI:** 10.3390/ijerph182010811

**Published:** 2021-10-14

**Authors:** Duc-Khanh Nguyen, Chung-Hsien Lan, Chien-Lung Chan

**Affiliations:** 1Department of Information Management, Yuan Ze University, Taoyuan 32003, Taiwan; ndkhanh95@gmail.com; 2Department of Computer Science, Nanya Institute of Technology, Taoyuan 32003, Taiwan; chlan@nanya.edu.tw; 3Innovation Center for Big Data and Digital Convergence, Yuan Ze University, Taoyuan 32003, Taiwan; 4ZDT Group—Yuan Ze University Joint R&D Center for Big Data, Taoyuan 32003, Taiwan

**Keywords:** healthcare, prediction, deep learning, deep ensemble learning approaches

## Abstract

With the development of information and technology, especially with the boom in big data, healthcare support systems are becoming much better. Patient data can be collected, retrieved, and stored in real time. These data are valuable and meaningful for monitoring, diagnosing, and further applications in data analysis and decision-making. Essentially, the data can be divided into three types, namely, statistical, image-based, and sequential data. Each type has a different method of retrieval, processing, and deployment. Additionally, the application of machine learning (ML) and deep learning (DL) in healthcare support systems is growing more rapidly than ever. Numerous high-performance architectures are proposed to optimize decision-making. As reliability and stability are the most important factors in the healthcare support system, enhancing the predicted performance and maintaining the stability of the model are always the top priority. The main idea of our study comes from ensemble techniques. Numerous studies and data science competitions show that by combining several weak models into one, ensemble models can attain outstanding performance and reliability. We propose three deep ensemble learning (DEL) approaches, each with stable and reliable performance, that are workable on the above-mentioned data types. These are deep-stacked generalization ensemble learning, gradient deep learning boosting, and deep aggregation learning. The experiment results show that our proposed approaches achieve more vigorous and reliable performance than traditional ML and DL techniques on statistical, image-based, and sequential benchmark datasets. In particular, on the Heart Disease UCI dataset, representing the statistical type, the gradient deep learning boosting approach dominates the others with accuracy, recall, F1-score, Matthews correlation coefficient, and area under the curve values of 0.87, 0.81, 0.83, 0.73, and 0.91, respectively. On the X-ray dataset, representing the image-based type, the deep aggregation learning approach shows the highest performance with values of 0.91, 0.97, 0.93, 0.80, and 0.94, respectively. On the Depresjon dataset, representing the sequence type, the deep-stacked generalization ensemble learning approach outperforms the others with values of 0.91, 0.84, 0.86, 0.8, and 0.94, respectively. Overall, we conclude that applying DL models using our proposed approaches is a promising method for the healthcare support system to enhance prediction and diagnosis performance. Furthermore, our study reveals that these approaches are flexible and easy to apply to achieve optimal performance.

## 1. Introduction

Healthcare support systems are entering a new era with the backing of information technology in the boom of big data and the Internet of Things. Following this context, artificial intelligence, especially deep learning (DL), is growing as a remarkable method of making predictions and classifications. As a result, it is becoming a reliable source for healthcare workers to make the final diagnosis and prescribe suitable treatments for their patients [1,2]. The applications of DL in healthcare are wide and abundant. Furthermore, improvements in the concepts and architectures of DL with outperformance in particular fields of healthcare are being achieved. Hence, the applications of DL in healthcare still have room for innovative research.

In data analysis, especially in the healthcare domain, there are three typical types of data, namely, statistical data, image-based data, and sequential data. Examples of these types are the records of patient information (diagnosis, operations, therapy, laboratory tests, symptoms, etc.) representing statistical data, CT images representing image-based data, and the recorded human index signals on wearable devices representing sequential data. These three types of data have different methods of retrieval, processing, and deployment. Statistical data essentially consist of independent variables or features and dependent variables or labels. Each observation usually has its own information on every feature and is represented by a row in a data table. Unlike statistical data, image-based data are made up of pixels arranged side by side. Each image has three dimensions, height, width, and depth, where height is the number of pixels lined up by height, width is the number of pixels lined up by width, and depth is the number of pixel layers stacked on top of each other. Sequential data are a special type of statistical data recorded in sequence over a period of time. The values of data in this sequence are relevant to each other. Due to the differences in the architecture of these three data types, particular prediction and classification models are deployed for the corresponding requirements. This leads to difficulty in implementing well-performing models in all data types.

Furthermore, in the healthcare support system, reaching the optimal performance is always the top priority for prediction and classification, where the decision of the healthcare workers needs to be accurate and tailored to each patient. Under these circumstances, ensemble techniques are one of the best choices [3]. Essentially, the ensemble learning technique combines many similar or different weak prediction models into a robust model. In other words, the technique is able to reduce variance and prevent overfitting phenomena in the training process [4,5]. As a result, it improves the accuracy and stability of the prediction model in classification and regression tasks. By incorporating DL models with ensemble learning techniques in this study, we propose three approaches collectively known as deep ensemble learning: deep-stacked generalization ensemble learning, gradient deep learning boosting, and Deep aggregation learning. In other words, by replacing the core learning units of the corresponding ensemble technique with suitable DL models, our proposed method can perform well with higher efficiency on all three data types.

The rest of this study is organized as follows. In Section 2, we review several recent studies on the application of DL in the healthcare domain and consider the way they were deployed on the three data types, as well as the proposed deep ensemble learning models. In Section 3, we introduce our benchmark datasets, which represent the three data types and provide a clear explanation, including some descriptive statistics. We also introduce the concepts of DL with their corresponding data type. Later, we present the proposed deep ensemble learning approaches and their architectures. Section 4 presents our deployed methods. In detail, we propose a scratch workflow for our proposed approaches and the step-by-step deployment of our experiment. Section 5 shows our experimental results. We also compare our proposed approaches to other individual DL and traditional machine learning techniques having similar ensemble concepts. Finally, our discussion and conclusions are presented in Section 6.

## 2. Literature Review

Currently, with the development of artificial intelligence, variants and their applications of DL in healthcare are growing rapidly [6,7]. They show high performance in early diagnosis and prediction, and as a result, can become a reliable reference source for healthcare workers to make the final decision for their patients [8,9]. Furthermore, these DL-based applications are conducted in different ways according to their data type.

In particular, for the statistical data type, the multilayer perceptron is the most deployed. In the study by Hosseinzadeh et al., a fine-tuned multiple multilayer perceptron neural network was deployed for thyroid disease diagnosis on the Internet of Medical Things (IoMT) [10]. In their application on IoMT systems, a final accuracy of 99% was obtained. According to the results, the proposed system has a remarkable capability to deal with overfitting during the training process. Abdar et al. used a multilayer perceptron neural network (MLP) to enhance the diagnosis of liver disease [11]. The results indicate that their MLP network, which is based on decision tree algorithms, achieved a higher accuracy of 14.57% than other traditional ML methods. In addition, Verma et al. conducted an MLP to classify coronary artery disease (CAD) in the CAD-56 and CAD-59 datasets with high accuracy of 86.47% and 98.35%, respectively.

Being different from the statistical data type, the image-based data type has its own deployment of DL concepts and applications. In detail, Stephen et al. proposed a convolutional neural network model trained from scratch to classify and detect the presence of pneumonia from chest X-ray image samples [12]. This model can help mitigate the reliability and interpretability challenges often faced when dealing with medical imagery. The result shows that their proposed method achieves a remarkable validation accuracy of 94%. Another study by Babukarthik et al. proposed a genetic deep learning convolutional neural network model to identify pneumonia due to COVID-19 using lung X-ray images [13]. This method performs better compared to other transfer learning techniques with a classification accuracy of 98.84%, a precision of 93%, a recall of 100%, and a specificity of 97% in COVID-19 prediction. Moreover, for medical image segmentation, Teng et al. proposed a deep multiscale convolutional neural network model [14]. Their proposed method is not only able to boost the segmentation accuracy but is also robust compared with other segmentation methods. For staging of liver fibrosis, Yasaka et al. proposed a deep convolutional neural network approach using gadoxetic acid-enhanced hepatobiliary phase magnetic resonance images [15]. They stated that this method exhibited high diagnostic performance in the staging of liver fibrosis with the highest areas under the ROC curve of 0.85. Another study by Yasaka et al. deployed a convolutional neural network for the differentiation of liver masses with dynamic contrast-enhanced computed tomography [16]. They conducted experiments on CT image sets of liver masses over three phases (non-contrast-agent enhanced, arterial, and delayed). The masses were diagnosed according to five categories. The experimental results showed that their proposed approach achieves high performance with a median accuracy and AUC of 84% and 0.92, respectively. The commonality of these studies is that they all use the convolutional neural network for prediction and classification.

On the other hand, the sequential data type also has its own deployment of DL concepts and applications. Specifically, in order to detect Congestive Heart Failure (CHF) via Short-Term RR Intervals, Wang et al. proposed an LSTM-based deep network approach [17]. This method aims to help clinicians monitor CHF out-patients and also make sense of HRV signals. The proposed model’s verification using the Beth Israel Deaconess Medical Center-CHF dataset achieves 99.22% of the highest accuracy. An end-to-end system is also provided to detect CHF using a short-term assessment of heartbeat. Another study from Yin et al. presented the MC-LSTM in order to warn for health emergencies with a low latency output and action and early detection capabilities [18]. They stated that their proposed system can achieve high performance with early detection features on trimmed and untrimmed cases of general-purpose and medical-specific datasets. Chen et al. proposed an attention-based bidirectional LSTM (Att-BiLSTM) method with service robots to classify outpatient categories according to textual content; hence, the robot can tell them which clinic they should register with [19]. The Att-BiLSTM developed system can achieve an accuracy of 96% correct responses to patients’ queries. Consequently, it opens an innovative approach to improve artificial intelligence in healthcare applications. As mentioned in the above studies, they shared the commonality of using recurrent neural networks (RNNs) for prediction and classification. In fact, there are even some studies that showed that the convolution neural network can also be used for sequential data; however, only the RNN concepts for sequential data will be applied here.

In general, the above methods all use DL with different modifications to deal with the characteristics of data. However, they all have some limitations for running a model independently. Because the above methods all use DL but are applied on different datasets, for the convenience of comparison, we have divided them into three groups corresponding to the three different data types with three different DL concepts. Table 1 describes each disadvantage in detail for the respective data types.

In addition, the ensemble learning technique gains much reliability owing to its performance in combining multiple predictive and classification models into one strong model. Ensemble learning with the core learning unit as DL is an innovative and prospective method. Several studies implemented this combination and showed its high performance and reliability as a result. Suk et al. presented a deep ensemble sparse regression network model to diagnose brain diseases [20]. Specifically, they combined the two conceptually different methods of sparse regression and DL for diagnosis and prognosis of Alzheimer’s disease and mild cognitive impairment. Their experiments achieved the highest diagnostic accuracy in three classification tasks compared to other traditional ML and individual DL models. A study by An et al. proposed the concept of deep ensemble learning, named deep belief network, to classify Alzheimer’s disease [21]. In brief, their proposed model has two layers, namely, a voting layer and a stacking layer. In the voting layer, they trained two sparse autoencoders to reduce the bias and extract the most important features. In the stacking layer, they applied a deep belief network with a nonlinear feature weighted to rank the base classifiers that may impact conditional independence. They deployed this network as a meta classifier. In the experimental results, their proposed framework achieves a 4% better accuracy than the other six ensemble approaches. Shahin et al. proposed a straightforward concept of ensemble learning by combining two DL architectures, namely, ResNet50 and Inception v3 to classify seven different types of skin lesions [22]. The experimental validation results achieve an accurate classification with a validation assurance accuracy of up to 89.9%. By proposing a deep ensemble learning model, named the deep-stacked generalization ensemble learning method, Nguyen et al. once again showed the advantage of this approach compared to other individual DL models and traditional ML models in the diagnosis of clinical depression from wearable device data [23]. Their proposed method is based on the stacked generalization ensemble learning technique and uses DL models as the core learning unit. The result on the independent validation set showed outstanding performance compared to other corresponding individual DL models.

The above evidence proves that DL and deep ensemble learning are prospective approaches for enhancing prediction performance, thereby becoming reliable resources for physician decision-making as a result.

## 3. Materials and Proposed Approaches

In this section, we introduce our benchmark datasets that represent the three data types with a clear explanation including some descriptive statistics. We also introduce the concepts of DL with the corresponding data type. After that, we present our proposed deep ensemble learning approaches and their architectures.

### 3.1. Benchmark Materials and Their Specifications

To prove the performance of our proposed approaches, we conducted experiments on three open datasets corresponding to the three data types. They are the Heart Disease UCI (HDU) dataset representing the statistical data type, the X-ray dataset representing the image-based data type, and the Depresjon dataset representing the sequence data type [24,25,26]. The HDU is a classic statistical dataset provided by the UCI Machine Learning Repository. It has a total of 270 samples with 13 recorded human index attributes. The HDU contains 150 normal and 120 observations having heart disease samples. The X-ray dataset is collected from pediatric patients with retrospective cohorts of one to five years. The patients were graded by two expert physicians at the Guangzhou Women and Children’s Medical Center. The dataset has a total of 5856 samples with 1583 normal images and 4273 pneumonia-presence images. The Depresjon dataset belongs to the sequential data type, it contains the total gravitational acceleration measured by wearable devices on 23 depressed (unipolar and bipolar) patients and 32 healthy control people at one-minute intervals. On this dataset, we applied a seven-day forward moving window with seven days of recorded data at a one-minute interval to generate a total of 814 samples. Section 4.1 describes the way these samples were generated. In all, 267 samples belong to the group of depressed patients and 547 samples belong to the control group of healthy people. The detailed description of these datasets is presented in Table 2.

### 3.2. Deep Learning and Its Concepts

Deep learning is a subclass of ML. The main idea of DL comes from the multilayer perceptron, which consists of nodes with corresponding activation functions, layers, and connections. In the classification task, the information from nodes belonging to the previous layer are initially fed to the forward layers via connections. The nodes belonging to the forward layer then combine all the information and later yield the data by using the activation function to feed to the forward nodes, and so on. In the end, the last layer contains several nodes, representing the number of classes that need to be classified, which will determine the probability of the corresponding predicted class (Figure 1).

The convolutional neural network (CNN) is another DL concept that is popular for image-based classification tasks. The adjacent convolution and pooling layers are the main components of this concept, and a fully connected layer at the end of the architecture yields the final prediction or classification. In particular, the convolution layers extract the features from previous layers using a sliding-window filter. The output of the convolution layers will be reduced in dimension by the pooling layers. The output of these layers are the so-called feature maps. In the end, the feature map of the last layer will be flattened and fed into a fully connected layer to make the prediction (Figure 2).

The recurrent neural network (RNN) is another neural network concept used to deal with sequential data. It uses the sequence of previous information to make an upcoming prediction or forecast. Essentially, the main components of this concept are the neural network cells, the sequence of input, and the output over iterations (Figure 3). The output of this cell later becomes its own input at the next state in a sequence. The way information is delivered in this concept accounts for the name, recurrent. A well-known RNN architecture, named long short-term memory (LSTM), is deployed in our experiments. The main components of LSTM are gates with different corresponding activations: forget gate, input gate, and output gate. These gates keep, store, or allow information go through to the next state. Consequently, this model is able to process a long sequence of data with the extracted and suitable-context information.

In this study, the core learning unit (CLU) of our proposed approaches is a simple DL based on the multilayer perceptron model for the statistical dataset, the ResNet50V2 models belonging to the convolution neural network, and the GRU models with an attention mechanism belonging to the recurrent neural network, which are deployed for the image-based and sequential datasets, respectively.

### 3.3. Proposed Deep Ensemble Learning Approaches

Ensemble learning essentially is a technique that combines many different or similar classifier models into one, and later makes the final prediction based on the members’ prediction results. The ensemble learning technique is able to improve the stability and accuracy of machine learning algorithms used in classification and regression by reducing variance and avoiding overfitting [27]. By taking advantage of these strengths, we propose three deep ensemble l (DEL) methods based on three corresponding ensemble learning techniques. They are deep-stacked generalization ensemble learning, gradient deep learning boosting, and deep aggregated learning. The main idea of our proposed approaches is to replace the core learning unit (CLU), the sub-classifier of ensemble models for traditional architectures with suitable DL models. As a result, our proposed approaches are deployed on the three data types. A further description of these proposed methods is presented below.

Deep aggregated learning (DAL) is the most straightforward deployment technique in our proposed deep ensemble learning family. The background of this method comes from the bootstrap aggregation technique [28]. By using this technique as the backbone and the deep neural networks (DNNs) as the sub-prediction model or CLU, we propose the DAL. The workflow of our proposed DAL method is presented as follows. First, the dataset is split into multiple subsets, and then multiple different or similar DNNs are trained from their corresponding subset. The output of every single DNN later is aggregated using a statistic, such as average, mode, etc., to yield the final prediction (Figure 4). In this proposed approach, the mean is deployed as an aggregated function. The formula of this approach is presented in Equation (1).
(1)$$P=\frac{\sum_{i=1}^{n}\left(P_{i}\right)}{n}$$
where:*P*: The final prediction.*P_i_*: The prediction of *i*^th^ model.*n*: The number of member model.

In gradient deep learning boosting (GDLB), the main idea of this proposed approach comes from the combination of the gradient boosting technique and the DNNs. In particular, the main component of a gradient boosting architecture is the booster or the classifiers and their residual output. The DNNs are deployed as the boosters following the gradient boosting backbone. This architecture effort is to boost the weak boosters into a high-performing classifier model. Specifically, the dataset is fed into every single booster. In the beginning, the first booster yields the mean value of the target labels, which is the so-called basement booster. The output of the first booster is the first prediction in the sequence of the gradient boosting architecture. After that, the residuals between this output and the real label turn into the next booster’s target value for training. In other words, the second booster will try to predict the residuals made from the previous booster, instead of predicting the real labels. The output of the second booster will be multiplied with a random learning rate ranging from 0 to 1 and previous predictions, including the basement prediction, will be added. This summation is the prediction of the second booster. Subsequently, the residual between the second prediction and the real labels becomes the target value of the third booster (Equation (2)). As a result, the residuals are minimized over the number of iterations. In the deployment of this proposed approach, we replace the traditional boosters with the CLU of DL models corresponding to the data type (Figure 5).
(2)$$P=P_{base}+\lambda \;\left[\overset{n}{\underset{i=1}{\sum }}\left(T_{i}-O_{i}\right)\right]$$
where:*P*: The prediction of the model with *n* booster models.*P_base_:* The 1^st^ prediction with default value.*λ*: The random learning rate ranging from 0 to 1.*T_i_*: The target that needs to be predicted by *i*^th^ model.*O_i_*: The output of the *i*^th^ model.

The deep-stacked generalization ensemble learning (DeSGEL) proposed approach is based on the stacked generalization concept [29]. In detail, there are two levels in making the prediction. In the lower level, called level 0, the data are fed into several similar classifier models for training. The information that goes through these models has been bias-reduced and is also extracted and compressed to obtain the most valuable features. The output of each classifier will then be combined into new so-called metadata. The metadata will be sent to the higher level, called level 1. At this level, the data will be fed into the last classifier to make the final prediction. This model is able to weigh the output of every single well-trained model from level 0 differently. As a result, it potentially gives better performance in the final prediction. In this proposed approach, we deploy the CLU of the DL models corresponding to the data type (Figure 6). The formula of this proposed approach is presented in Equation (3).
(3)$$P_{n}=f\left\{\overset{n}{\underset{i=1}{⋃}}P_{i}\right\}$$
where:*P_n_*: The prediction of model with *n* booster models.*f*: The final classifier function*P_i_*: The prediction of *i*^th^ classifier.*n*: The number of member models.

## 4. Methods

In this study, to demonstrate the adaptability and outstanding performance of our proposed approaches, we conducted the experiments on three open datasets corresponding to statistical, image-based, and sequential data types. In addition, we also compared our proposed approaches with the corresponding single DL model and other ML models having the same methodology, namely, random forest, gradient boosting, and stacking classifier. All of our deployments comparing ML models were implemented by the Scikit-learn open-source library. The hyperparameter configurations of these deployed models will be presented in Section 4.2. In general, we proceeded with our experiments in three phases (Figure 7). In Phase 1, after the data reading step, we conducted preprocessing steps on the different datasets with different methods to transform the data into suitable formats. After that, in Phase 2, we built our proposed approaches and deployed other traditional ML models corresponding to the dataset with the different data types. The hyperparameters of the deployed models were also tuned in this phase. Finally, in Phase 3, we evaluated the deployed models as well as our proposed approaches and conducted a performance comparison among the models.

Our experiments were conducted with Anaconda software v1.10.0, Python v3.7, with an open-source packet library of ML and DL, named Scikit-learn v0.24.1 and Tensorflow v2.1.0 [30,31,32]. All experimental processes are conducted on a single machine having specifications as Intel(R) Xeon(R) E3-1220 v6 of CPU, 40GB DDR4 of RAM, and GeForce GTX 1080 Ti 11GB of GPU memory. The detailed deployment of our workflow is presented in sub-sections below.

### 4.1. Preprocessing (Phase 1)

With statistical datasets, there are issues such as missing data, irrelevant features, and data with different-scale features. In order to deal with these matters, we first applied the imputation method to fulfill the missing values with the mean of the non-missing values of the corresponding feature. After that, we sliced the dataset into training, validating, and testing sets into 10-fold training and validating sets with the ratio of 80:4:16. Subsequently, we used the standard scaler method to make the feature on the same scale (Equation (4)). In detail, the mean and standard deviation of the scaler were computed on the training set; then, we performed standardization by centering and scaling on the training, validating, and testing sets. In the end, a random forest (RF) was deployed to rank the importance of the features. This RF contains 100 decision trees, with each tree built by a random subset with two nodes. The RF is able to divide the dataset into two clusters, in which the samples in the same cluster are similar to each other and different from the samples belonging to the other clusters.
(4)$$x_{scaled}=\frac{x-\mu }{\sigma }$$
where:μ: Mean.σ: Standard deviation.

The samples for the image-based datasets were originally in different sizes. Therefore, to feed the image into the training models, we first resized each image to 200 × 200 × 1 where 200 is the value of height and width, and 1 is the depth of the image. In addition, the traditional ML models that we deploy in our study (e.g., random forest, gradient boosting, and stacking classifier) are not suitable for the image-based dataset. Hence, before deploying these ML models, we transformed these images into statistics-based data using the Convolutional Auto-Encoder (CAE), its architecture is presented in Table A1. Generally, the CAE is used for image de-noising by extracting the most important features of the image and then trying to generate similar images based on the extracted features on the output layer. In detail, the main components of CAE are encoding convolutional layers, a compressed convolutional layer, and decoding convolutional layers. In our experiments, after splitting the data into training, validation, and testing sets, we first trained our CAE model on the training set. The weight of a well-trained model is retained. After that, we extracted the information from the compressed convolutional layer and fed it into the traditional gradient boosting and random forest for training and making predictions. We performed the same steps on the testing set; however, instead of training the DL model again, we fed the data into the well-trained models with the frozen weights, and the output of the compressed convolutional layer was fed into the traditional gradient boosting, random forest, and stacking classifier.

Our sequential dataset, Depresjon, also contains missing values due to subjective factors from the participants (e.g., participants took off their device when sleeping or taking a shower, or when the device needed to be charged, etc.). To deal with the missing data, we applied the mean imputation method. However, instead of filling the missing values by the mean of the corresponding non-missing values for the corresponding features, we inserted the mean of non-missing values coming from the same participant and the same missing time of day. We also applied the seven-day forward moving window method to generate a seven-day recorded sample (Figure 8). In other words, each sample was generated by a window covering the information of every minute of seven consecutive days for a total of 10,080 recorded minutes. After generating a new sample, the starting point of this window was moved to the next day. In total, 814 samples were generated including normal samples and depression-appearance samples, and each sample contained 1440 recorded values in sequence.

### 4.2. Building Models, Starting Training and Tuning (Phase 2)

Our proposed models are built from three ensemble learning approaches, namely DeSGEL, GDLB, and DAL. In general, the CLUs differ among the approaches and data types. Specifically, for the statistical dataset, the CLUs are the simple fine-tuned DNN models of which architectures are presented in Table A2. For the image-based dataset, the CLUs are the well-known architecture of the CNN concept with a combination of input, convolutional, pooling, and output layers named ResNet50V2 appended to a flat feedforward neural network that makes the final prediction. The architecture of this model is presented in Table A3. For the sequential dataset, the deployed CLUs consist of a Long-Short Term Memory (LSTM), which is the architecture of the RNN concept, and a modified simple DNN at the end. Their architectures are presented in Table A4. In addition, for the DeSGEL approach, the level 1 classifiers are the same in all datasets (Table A5).

In this study, we used the training set for tuning the hyperparameters and the independent testing set for evaluating and comparing. In our experiment, we considered two hyperparameters, namely, the number of sub-models and the number of epochs inside the sub-models. The number of sub-models ranged from 5 to 50, and the number of epochs ranged from 50 to 500, with a 50-step increment for each proposed approach. The best hyperparameter sets for each data type are presented in Table A6. Furthermore, to enhance the training performance, we also applied some tricky techniques, namely, changing the learning rate and saving the network’s weight according to the best performance over the epoch. All of these functions are provided in the Keras open-source library. Specifically, we used the LearningRateScheduler function to change the learning rate (λ) every 50 epochs. The updated λ was then calculated by the formula λ = λ − λ * 0.1. In addition, we applied the ModelCheckpoint function to save the weight of the network according to the loss-of-validation set when it reached the minimum value. The optimization algorithm for all DAL and DeSGEL sub-models was fixed with the Adam optimizer with a binary cross-entropy loss function. However, the GDLB used mean squared error as the loss function and stochastic gradient descent as the optimizer function because the predicted values of the CLU are regression instead of binary class.

Additionally, traditional ML and DL models, random forest, gradient boosting, stacking classifier, and DNN were also deployed to compare with the corresponding proposed DEL approaches. Their best hyperparameter sets are presented in Table A6. The stacking classifier, on the other hand, is a technique that combines many ML methods as low-level models. The higher-level model is trained based on the output of the lower-level models and yields the final prediction. Following this structure, in our experiment, we used random forest and gradient boosting with the same-mentioned hyperparameter sets as the low-level classifiers and a logistic regression with default parameters as the higher-level classifier. Furthermore, the DNN was deployed on different data types following different concepts. The architectures of these DL networks are similar to the CLUs of our proposed approaches for the corresponding dataset. A summary of these hyperparameter sets is presented in Table A6.

### 4.3. Evaluating Models (Phase 3)

In this study, the predictions were evaluated using a confusion matrix and its derived metrics (Figure 9). Essentially, this matrix is made up of four components: true negative (TN), true positive (TP), false negative (FN), and false positive (FP). The true negative samples that were predicted as negative are represented by TN. The true positive samples that are predicted as positive are represented by TP. The true positive samples that are predicted as negative are represented by FN. The true negative samples that are predicted as positive are represented by FP.

From the confusion matrix, the accuracy, precision, recall, F1-score, Matthews correlation coefficient (MCC), and area under the curve (AUC) metrics (Equations (5)–(9)) were derived, in which the AUC represents the ability to distinguish among classes. Its value ranges from 0 to 1, where 1 is the best discrimination of the model. Formulas for the other metrics are presented below.
(5)$$\mathrm{Accuracy}=\frac{\mathrm{TP}+\mathrm{TN}}{\mathrm{TP}+\mathrm{FP}+\mathrm{FN}+\mathrm{TN}}$$
(6)$$\mathrm{Precision}=\frac{\mathrm{TP}}{\mathrm{TP}+\mathrm{FP}}$$
(7)$$\mathrm{Recall}=\frac{\mathrm{TP}}{\mathrm{TP}+\mathrm{FN}}$$
(8)$$F1\;\mathrm{Score}\;=\frac{2\;\times \;\mathrm{Precision}\;\times \;\mathrm{Recall}}{\mathrm{Precision}+\;\mathrm{Recall}}$$
(9)$$\mathrm{MCC}\;=\frac{\mathrm{TP}\;\times \;\mathrm{TN}-\mathrm{FP}\;\times \;\mathrm{FN}}{\sqrt[{}]{\left(\mathrm{TP}+\mathrm{FP}\right)\left(\mathrm{TP}+\mathrm{FN}\right)\left(\mathrm{TN}+\mathrm{FP}\right)\left(\mathrm{TN}+\mathrm{FN}\right)}}$$

## 5. Results

We conducted the experiments on three open datasets representing three kinds of data types with seven models for each. In total, 21 trained-by-task ML and DL models were deployed. Overall, the proposed DEL approaches show the outstanding outperformance of other ML and individual DL models (Table 3).

Specifically, on the HDU statistical dataset, our proposed DEL family of models almost dominates the corresponding traditional ML and DL groups by 3 to 7% in all evaluation metrics. There are different evaluation performances among the proposed approaches; however, the gaps are not too great. These approaches share the highest accuracy, F1-score and MCC of 0.87, 0.83 and 0.73, respectively. The GDLB with the CLU of DNN outperforms others with the highest recall and AUC of 0.81 and 0.91. Even the individual DL model shows the highest precision of 0.90, but only that metric is insufficient for a good model.

Furthermore, on the image-based dataset named X-ray, our proposed approaches also show outstanding performance on the average evaluation metrics with values of 0.88, 0.88, 0.94, 0.90, 0.74 and 0.90, respectively. In particular, the DAL reaches the highest performance in accuracy, precision, F1-score, MCC, and AUC of 0.91, 0.90, 0.93, 0.80 and 0.94, respectively. The DeSGEL obtained the highest recall of 0.98. However, the proposed GDLB approach shows poor performance when compared with the in-family models and also with the ML and individual DL group. The ML and DL models also achieve the highest recall of 0.98.

Finally, on the sequential dataset, Depresjon, once again, the family of proposed approaches completely outperforms the corresponding ML and DL groups by 3% to even 13% on the average evaluation metrics. In particular, the DeSGEL approach dominates others in all evaluation metrics on accuracy, precision, recall, F1-score, MCC, and AUC of 0.91, 0.88, 0.84, 0.86, 0.80, and 0.94, respectively.

In summary, among the proposed approaches, the GDLB with DNN CLU is slightly better than the others on the statistical datasets. On the other hand, the DAL approach shows the best performance over other approaches in the DEL family supporting CNN CLU (ResNet50V2). The DeSGEL with GRU combined with the attention mechanism outperforms others in the sequence dataset.

The comparison of our proposed approaches with other DL and ML models is conducted by deploying all models on one single computer with the specification mentioned above. Table A7 presents the results of the training and classification processes in terms of time and hardware-resource consumption. In conclusion, the proposed DEL approaches combine multiple models in one runtime; hence, this processing step consumed more hardware resources and running time compared with other individual DL and ML methods. In fact, the CLUs of the proposed approaches are trained and used to predict on GPU but traditional ML methods are not; hence, we have not included GPU as a hardware-resource consumption factor in the comparison.

## 6. Conclusions

Healthcare support systems are improving with the support of artificial intelligence, especially DL. Several innovative studies applying DL in early prediction and diagnosis have given good performance. However, machine- and software-based diagnosis and prediction in healthcare are still under development. In this study, we propose DEL approaches to enhance the accuracy and reliability of these predictions. The results show that the approaches in the DEL family dominate over all other individual deployed models in performance on all types of dataset. According to the experiment results, using DL as the CLU, the GDLB approach is suitable for the statistical data type, the DAL approach is advisable for the image-based data type, and the DeSGEL approach is recommended for the sequential data type. In addition, the workflow and deployment method also play an important role in the implementation process. Our study presents a general and particular process from scratch on different types of data that do not require much technical domain knowledge to implement. In addition, by replacing the suitable CLUs of DL for each data type, the comparison between our proposed approaches and the corresponding individual DL and ML models with similar architecture shows that the proposed approaches outperform the others overall. Furthermore, the proposed approaches are capable of tuning the hyperparameters such as learning rate, activation function, optimization function, etc., to improve the learning process. In this study, we deploy the same architecture of CLUs. However, our proposed approaches are flexible and easy to apply to state-of-the-art models by replacing the CLU with these DL models to achieve optimal performance in future applications. There are some limitations to our approaches, such as the training time and hardware resource consumption due to combining several DL models as the CLUs. However, this is a worthwhile trade-off as IT hardware resources become more and more affordable, and the requirements of accuracy and reliability are always the top priority in the healthcare field. Another important factor in the healthcare support systems is the explainable feature affecting the prediction, but in these approaches, we did not show a way to deal with this issue. This is also a general drawback of DL approaches. The classification task is the main focus of our study. However, there are further tasks that our approaches will be able to manage, such as segmentation, forecasting, etc. In conclusion, we discovered that applying DEL models using our proposed approaches is a prospective process in the healthcare support system in order to enhance prediction and diagnosing performance.

## Figures and Tables

**Figure 1 ijerph-18-10811-f001:**
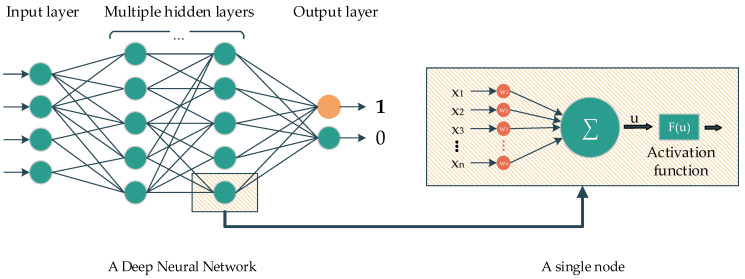
A Deep neural network and its elements.

**Figure 2 ijerph-18-10811-f002:**
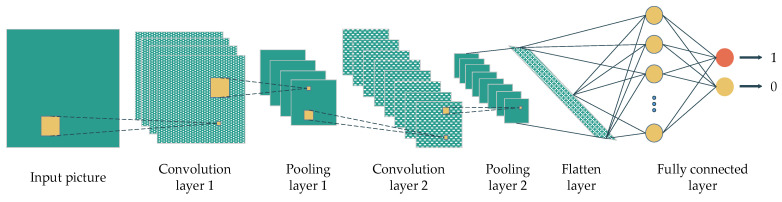
A convolution neural network’s architecture with the main components, namely convolution layers, pooling layers, flatten layer and fully connected layers.

**Figure 3 ijerph-18-10811-f003:**
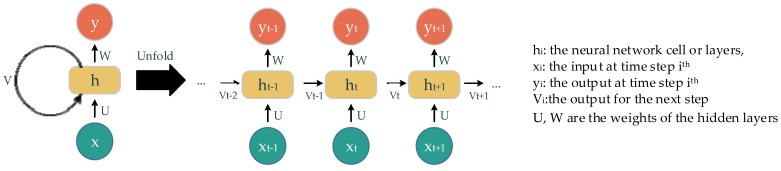
The recurrent neural network and its un-fold state.

**Figure 4 ijerph-18-10811-f004:**
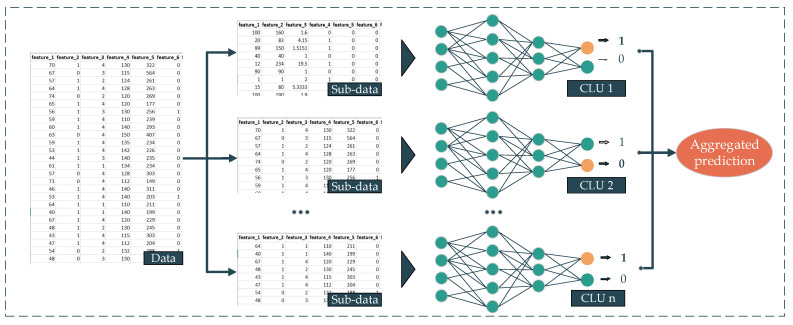
Deep aggregate learning method.

**Figure 5 ijerph-18-10811-f005:**
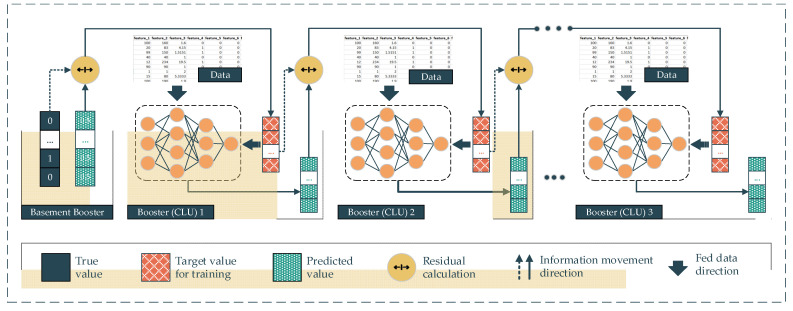
Gradient deep learning boosting method.

**Figure 6 ijerph-18-10811-f006:**
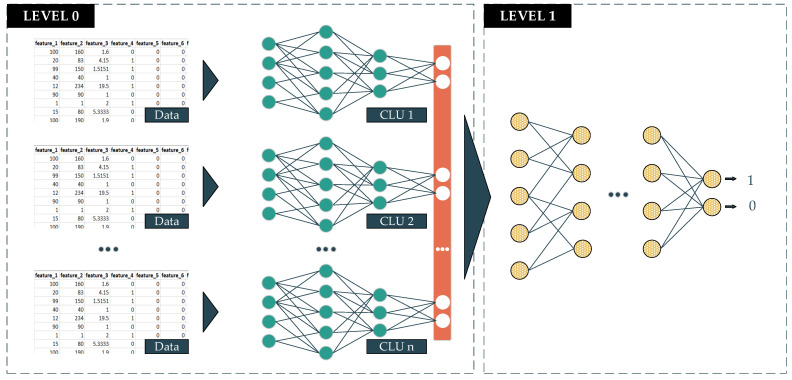
Deep Stacked Generalization Ensemble Learning method.

**Figure 7 ijerph-18-10811-f007:**
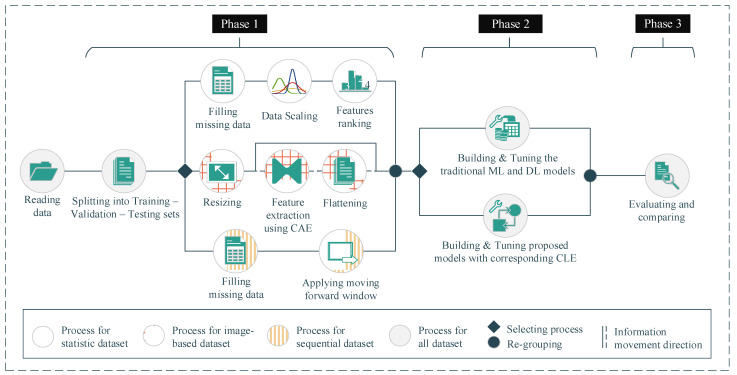
Experimental workflow. (Phase 1: preprocessing; Phase 2: building, training, and tuning the models; Phase 3: evaluating the models).

**Figure 8 ijerph-18-10811-f008:**
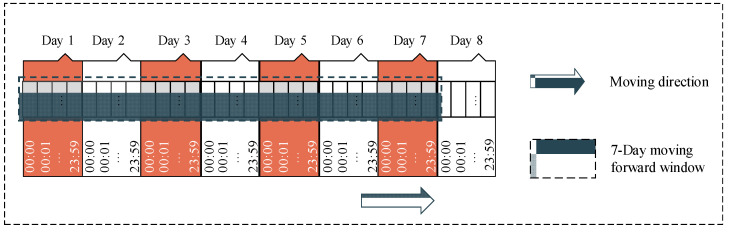
The seven-day forward moving window for sequence data with a length of 10,080 where 10,080 is the total number of minutes in seven days.

**Figure 9 ijerph-18-10811-f009:**
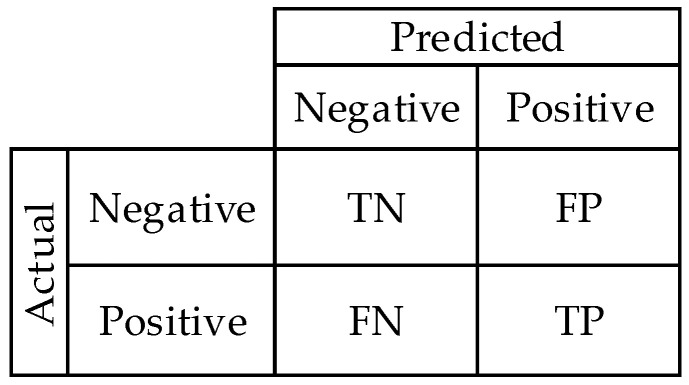
Confusion matrix.

**Table 1 ijerph-18-10811-t001:** The particular and common disadvantages of mentioned methods.

Existing DL ^1^ Methods	Particular Disadvantages	Common Disadvantages
DL methods on statistical data type (MLP ^2^)	The individual MLP are sensitive to the initial randomization of their weight matrices. The order in which we submit training data to a neural network affects the result. MLP often tends to end up in local minima of their loss function.	They become useless on imbalanced datasets. They require a large dataset to process and train the neural network to get good results. Each model is effective in solving just one particular class of problems, not all. The weight scale is faster than linearly when the network increases in size. The models are not all equal. Different model architectures may solve the same problem with comparable accuracy while requiring a significantly different amount of computation. Overfitting happens easily when the network is trained so many times
DL methods on image-based data type (CNN ^3^)	The individual CNNs lack the ability to be spatially invariant to the input data.	
DL methods on sequential data type (RNN ^4^)	The individual RNN models usually have vanishing gradients and exploding problems. Network stability is often difficult to ascertain due to the nonlinear nature of the unit activation output characteristics and the weight adjustment strategies	

^1^ Deep learning, ^2^ Multilayer perceptron, ^3^ Convolutional neural network, ^4^ Recurrent neural network.

**Table 2 ijerph-18-10811-t002:** Datasets description.

Dataset	Variable #	Sample #	[Normal/Disease] Ratio ^1^	Data Type
HDU ^2^	13	270	1.25	Statistical
X-ray	N/A	5856	0.37	Image-based
Depresjon	1	814	2.05	Sequential

^1^ Approximate ratio, ^2^ Heart Disease UCI dataset.

**Table 3 ijerph-18-10811-t003:** Comparison Results.

**Heart Disease UCI (Statistical Dataset)**						
**Model**	**Accuracy**	**Precision**	**Recall**	**F-Score**	**MCC ^1^**	**AUC ^2^**
DeSGEL ^3^ + DNN ^4^ CLU ^5^	**0.87** ± 0.01 *****	0.88 ± 0.01	0.78 ± 0.01	**0.83** ± 0.01	**0.73** ± 0.02	0.90 ± 0.00
GDLB ^6^ + DNN CLU	**0.87** ± 0.02	0.86 ± 0.02	**0.81** ± 0.04	**0.83** ± 0.02	**0.73** ± 0.03	**0.91** ± 0.01
DAL ^7^ + DNN CLU	**0.87** ± 0.01	0.88 ± 0.05	0.79 ± 0.04	**0.83** ± 0.02	**0.73** ± 0.03	0.89 ± 0.01
Average of proposed approaches on mean values	0.87	0.87	0.79	0.83	0.73	0.90
DNN	0.86 ± 0.01	**0.90** ± 0.02	0.74 ± 0.04	0.81 ± 0.02	0.71 ± 0.03	0.87 ± 0.02
Gradient Boosting	0.75 ± 0.05	0.68 ± 0.07	0.75 ± 0.06	0.71 ± 0.04	0.49 ± 0.09	0.80 ± 0.05
Random Forest	0.85 ± 0.01	0.85 ± 0.05	0.78 ± 0.05	0.81 ± 0.01	0.70 ± 0.03	0.90 ± 0.01
Stack Ensemble	0.84 ± 0.03	0.82 ± 0.05	0.78 ± 0.11	0.79 ± 0.05	0.67 ± 0.06	0.89 ± 0.02
Average of traditional DL and ML on mean values	0.83	0.81	0.76	0.78	0.64	0.87
**X-ray (Image-Based Dataset)**						
**Model**	**Accuracy**	**Precision**	**Recall**	**F-Score**	**MCC**	**AUC**
DeSGEL + ResNet50V2 CLU	0.89 ± 0.01	0.86 ± 0.01	**0.98** ± 0.01	0.91 ± 0.00	0.76 ± 0.01	0.88 ± 0.02
GDLB + ResNet50V2 CLU	0.84 ± 0.03	0.87 ± 0.04	0.88 ± 0.06	0.87 ± 0.03	0.65 ± 0.06	0.87 ± 0.03
DAL + ResNet50V2 CLU	**0.91** ± 0.01	**0.90** ± 0.02	0.97 ± 0.01	**0.93** ± 0.01	**0.80** ± 0.02	**0.94** ± 0.01
Average of proposed approaches on mean values	0.88	0.88	0.94	0.90	0.74	0.90
CNN ^8^ (ResNet50V2)	0.88 ± 0.03	0.85 ± 0.03	**0.98** ± 0.00	0.91 ± 0.02	0.75 ± 0.06	0.91 ± 0.05
Gradient Boosting with CAE ^9^	0.85 ± 0.01	0.82 ± 0.01	0.97 ± 0.00	0.89 ± 0.00	0.68 ± 0.01	0.81 ± 0.01
Random Forest with CAE	0.86 ± 0.00	0.83 ± 0.00	**0.98** ± 0.00	0.90 ± 0.00	0.71 ± 0.01	0.82 ± 0.00
Stack Ensemble with CAE	0.85 ± 0.01	0.82 ± 0.01	**0.98** ± 0.00	0.89 ± 0.00	0.69 ± 0.01	0.81 ± 0.01
Average of traditional DL and ML on mean values	0.86	0.82	0.98	0.90	0.71	0.84
**Depresjon (Sequential Dataset)**						
**Model**	**Accuracy**	**Precision**	**Recall**	**F-score**	**MCC**	**AUC**
DeSGEL + GRU ^10^ with Attention CLU	**0.91** ± 0.02	**0.88** ± 0.04	**0.84** ± 0.04	**0.86** ± 0.04	**0.80** ± 0.06	**0.94** ± 0.03
GDLB + GRU with Attention CLU	0.88 ± 0.01	0.85 ± 0.04	0.76 ± 0.03	0.80 ± 0.01	0.72 ± 0.01	0.93 ± 0.00
DAL + GRU with Attention CLU	0.87 ± 0.02	**0.88** ± 0.04	0.78 ± 0.05	0.79 ± 0.03	0.70 ± 0.03	0.90 ± 0.02
Average of proposed approaches on mean values	0.89	0.87	0.79	0.82	0.74	0.92
RNN ^11^ (GRU with Attention)	0.84 ± 0.01	0.78 ± 0.08	0.71 ± 0.12	0.73 ± 0.04	0.63 ± 0.03	0.87 ± 0.02
Gradient Boosting	0.82 ± 0.01	0.72 ± 0.05	0.72 ± 0.09	0.71 ± 0.03	0.59 ± 0.03	0.86 ± 0.01
Random Forest	0.84 ± 0.01	0.73 ± 0.03	0.79 ± 0.07	0.76 ± 0.02	0.65 ± 0.03	0.89 ± 0.01
Stack Ensemble	0.84 ± 0.01	0.73 ± 0.04	0.82 ± 0.08	0.76 ± 0.03	0.65 ± 0.03	0.90 ± 0.01
Average of traditional DL and ML on mean values	0.84	0.74	0.76	0.74	0.63	0.88

^1^ Matthews correlation coefficient, ^2^ Area Under the Curve, ^3^ Deep Stacked Generalization Ensemble Learning, ^4^ Deep Neural Network, ^5^ Core Learning Unit, ^6^ Gradient Deep Learning Boosting, ^7^ Deep Aggregated Learning, ^8^ Convolution Neural Network; ^9^ Convolution Auto-Encoder; ^10^ Gate Recurrent Unit; ^11^ Recurrent Neural Network. The best performance values are bold. ***** Each experiment is repeated 10 times and average results are presented in format: m ± d, where m is the average and d is the standard deviation across the 10 experiments, on the best hyper-parameter sets.

## Data Availability

The Depresjon Dataset could be downloaded at https://datasets.simula.no//depresjon (accessed on 3 September 2021). The X-ray dataset could be downloaded at https://www.kaggle.com/paultimothymooney/chest-xray-pneumonia (accessed on 3 September 2021). The Heart Disease UCI could be downloaded at https://github.com/khanhdc/Deep-Ensemble-Leanring/blob/main/Dataset/heart.dat (accessed on 3 September 2021).

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
