# Peer review of "Deep Ensemble Learning Approaches in Healthcare to Enhance the Prediction and Diagnosing Performance: The Workflows, Deployments, and Surveys on the Statistical, Image-Based, and Sequential Datasets"

_ijerph, 2021, doi:10.3390/ijerph182010811_

Round 1

Reviewer 1 Report

Paragraph 4.1 contains descriptions of the methodology used in the experiments. We suggest to split “Experiments and results” into “Methods” and “Results” in separate paragraphs. The former can be moved to Paragraph 3 “Materials and Proposed approaches”.

The discussion does not place the results within the current knowledge of the corresponding field of research, although extensive review of the literature (non-systematic) has been conducted in Paragraph 2.

Author Response

we would like to thank all the reviewers for taking your precious time on reading and giving back the feedback. We would like to respond according to your comments. We have carefully considered the comments and tried our best to address every one of them. Please see the attachment

Reviewer 2 Report

This is an interesting paper, in which the Deep Ensemble Learning approaches were proposed to deal with three different types of data. Moreover, the DEL approaches proposed by the authors outperformed than traditional method. Basically, the topic is interesting and the study is well done. However, I have following few suggestions: 1. The article is good in content, only it’s a bit wordy. It’s better to cut it down. In addition, language problems also can be seen everywhere. 2. Many abbreviations are used, but some are not given when they appeared in the first place. My suggestion is “don’t use more than 5 abbreviations”, too many abbreviations will make the reading or reviewing difficulty. 3. On page 5, for the third data set, “applied a 7-Days moving average window with 7 days of recorded value every single minute to generate a total of 814 samples. In which, 267 samples are belonging to the Depressed class and 547 samples belonging to healthy control people.” As there are 23 depressed patients and 32 healthy control people, each depressed patient has 11.6 samples. How did this number be generated? 4. In the results part, you should give the running time for different model, and also the hardware resource.

Author Response

We would like to thank all the reviewers for taking your precious time on reading and giving back the feedback. We would like to respond according to your comments. We have carefully considered the comments and tried our best to address every one of them. Please see the attachment

Reviewer 3 Report

The authors explained how to gather, access, and store patient data at any time. These data are useful for monitoring, diagnosing, data analysis, and decision making. The data can be classified as statistical, image-based, or sequential. Each one retrieves, processes, and deploys data differently. However, the use of Machine Learning (ML) and Deep Learning (DL) in healthcare is fast increasing. Many high-performance decision-making architectures were proposed. The healthcare support system must be reliable and stable. So improving anticipated performance and keeping model stability is always the first goal. The authors developed three Deep Ensemble Learning (DEL) techniques that function reliably on all data types. Their study's fundamental notion is ensemble approaches. The ensemble models achieved remarkable performance with firmness and trustworthiness by integrating numerous weak models into one.

These include Deep Stacked Generalization Ensemble Learning (DeSGEL), Gradient Deep Learning Boosting (GDLB), and Deep Aggregation Learning (DAL) (DAL). The results demonstrated that their proposed techniques outperformed classical ML and DL on all statistical, image-based, and sequential benchmark datasets. AUC of 0.87, 0.81, 0.83, 0.73, 0.91 for the GDLB on the Heart disease UCI dataset. The DAL performed the best on the X-Ray dataset, with scores of 0.91, 0.97, 0.93, 0.80, and 0.94. The DeSGEL outscored others by 0.91, 0.84, 0.86, 0.8, and 0.94 on the Depression dataset reflecting sequence type. Using DL models to improve prediction and diagnosis performance is a prospective activity in the healthcare support system. Their tactics are also versatile and straightforward to apply to archive the best performance.

Please take some more parameters to evaluate the performance of your systems if possible, kindly add some references of 2020-21 related to recent work i.e. 

However, certain sections are to be fine-tuned.

A.

In the abstract, please give one line about the issues of the topic. 

  1. An Introduction, motivation of the paper is not clear.
  2. The disadvantages of the existing schemes must be discussed in tabular form. 
  3. The last paragraph of the Introduction must represent the structure of the paper in a more inspiring way.
  1.  

The structure of the Introduction must be like this:

  1. A brief discussion about the topic
  2. The problem of the topic
  3. Existing solutions of the topic along with the problems of the existing schemes/solutions.
  4. Brief details about the proposed scheme. Contributions (point-wise)
  5. Proper Structure of the paper
  6. Some parts of the Introduction must be therein literature survey portion (review portion).

In Literature Survey.

  1. This section must be powerful. Please include many existing schemes with their disadvantages.
  2. Each existing scheme can be discussed separately, one by one, with its disadvantages. Otherwise, some similar schemes can be given in a paragraph and so on.
  3. In this section, the discussion must be started from the First existing model of the mentioned problem, i.e. foremost model.

C.

Proposed Work:

  1. Rename this section as “Proposed Scheme”.
  2. This section must deal with the proposed scheme. Do not include any existing concept. A crucial current idea must be given in a different section, i.e. Preliminary Studies.
  3. At the start of this section, please include a paragraph to discuss the proposed scheme briefly.
  4. There must be no reference in the Section heading.
  5. It is difficult to understand how the proposed scheme does work.
  6. Give a diagram to depict the workflow of the proposed scheme.
  7. Please divide the entire proposed scheme into different subsections/phases and then discuss the concept under each subsection.
  8. Step wise discussion is preferable.
  9. There must be discussion regarding Experimental Environment in a separate subsection.
  10. The proposed scheme must be compared with at least three existing related schemes.
  11. There must be technical discussions regarding the results.

Conclusion:

  1. There must be maximum of two paragraphs in this section. The first paragraph is for briefly discussing the entire paper and the second paragraph is for discussing some future works.

Additional Comments:

  1. Short-form must be defined as properly when used for the first time.
  2. Never use I, we, you, our, etc. in a research article. Use one “Tense” to write the entire paper. “Present Tense” is preferable.
  3. There are many typos and grammatical mistakes in the entire paper. Many lines are not understandable. In a sentence, don’t use Capital Letter for a Word unnecessarily.
  4. All the figures, tables, equations and references must be cited in the text.
  5. .References must be cited sequentially starting from 1, then 2, then 3 and so on.
  6. Use Matlab or GNU Plot or Graper or any other to draw the graphs of results.
  7. Add a separate section as “Performance Analysis” to discuss results and discussions. This section must have three subsections: (i) Experimental Environment (ii) Details of Dataset, and (iii) Results and Discussions.

Also, add your published papers (if any) on the same topic.

Final Decision

As the work done is useful so recommends the manuscript to be accepted with minor revisions.

Author Response

(The authors gave the same response as above.)
